# Epidemiology and (Patho)Physiology of Folic Acid Supplement Use in Obese Women before and during Pregnancy

**DOI:** 10.3390/nu13020331

**Published:** 2021-01-23

**Authors:** Melissa van der Windt, Sam Schoenmakers, Bas van Rijn, Sander Galjaard, Régine Steegers-Theunissen, Lenie van Rossem

**Affiliations:** Department of Obstetrics and Gynecology, Erasmus MC, University Medical Center, 3000 CA Rotterdam, The Netherlands; m.vanderwindt@erasmusmc.nl (M.v.d.W.); s.schoenmakers@erasmusmc.nl (S.S.); b.vanrijn@erasmusmc.nl (B.v.R.); s.galjaard@erasmusmc.nl (S.G.); l.vanrossem@erasmusmc.nl (L.v.R.)

**Keywords:** obesity, folic acid supplement use, neural tube defects

## Abstract

Preconception folic acid supplement use is a well-known method of primary prevention of neural tube defects (NTDs). Obese women are at a higher risk for having a child with a NTD. As different international recommendations on folic acid supplement use for obese women before and during pregnancy exist, this narrative review provides an overview of epidemiology of folate deficiency in obese (pre)pregnant women, elaborates on potential mechanisms underlying folate deficiency, and discusses considerations for the usage of higher doses of folic acid supplements. Women with obesity more often suffer from an absolute folate deficiency, as they are less compliant to periconceptional folic acid supplement use recommendations. In addition, their dietary folate intake is limited due to an unbalanced diet (relative malnutrition). The association of obesity and NTDs also seems to be independent of folate intake, with studies suggesting an increased need of folate (relative deficiency) due to derangements involved in other pathways. The relative folate deficiency, as a result of an increased metabolic need for folate in obese women, can be due to: (1) low-grade chronic inflammation (2) insulin resistance, (3) inositol, and (4) dysbiotic gut microbiome, which plays a role in folate production and uptake. In all these pathways, the folate-dependent one-carbon metabolism is involved. In conclusion, scientific evidence of the involvement of several folate-related pathways implies to increase the recommended folic acid supplementation in obese women. However, the physiological uptake of synthetic folic acid is limited and side-effects of unmetabolized folic acid in mothers and offspring, in particular variations in epigenetic (re)programming with long-term health effects, cannot be excluded. Therefore, we emphasize on the urgent need for further research and preconception personalized counseling on folate status, lifestyle, and medical conditions.

## 1. Rationale

In order to prevent neural tube defects (NTDs) in offsprings, women are advised to take a 0.4 mg folic acid supplement from the moment they wish to get pregnant up until the first trimester of pregnancy [1]. This advice applies to all women, except for women with a history of a previous child with a NTD, who are advised to take a higher dose of 4–5 mg folic acid supplement [1]. 

A growing number of women is obese when trying to get pregnant, with an increased risk of having a child with a NTD [2,3]. Meta-analyses showed a dose-response association between maternal Body Mass Index (BMI) and NTDs, and the risk rapidly increased in women with a BMI ≥ 30 kg/m^2^ (Table 1) [4,5,6]. In addition, a BMI ≥ 30 kg/m^2^, defined as maternal obesity, is also associated with the severity of the NTD in the offspring [7,8].

Given the known association between inadequate maternal folate intake and NTD in offsprings, and the increased risk of NTDs in obese women, the question arises whether obese women more often have a folate deficiency [9]. There might be an absolute folate deficiency from diet (folate) due to a suboptimal intake that is associated with obesity, combined with the fact that obese women may be less compliant in taking supplements (folic acid) [10,11,12]. On the other hand, obese women can have a relative folate deficient status, caused by a state of chronic low-grade inflammation, which results in an increased metabolic need of folate. Importantly, studies have shown that obese women had an increased risk of NTDs, regardless of their folate intake [13,14]. There are no studies that have assessed whether a high dose of folic acid results in less NTD pregnancies in obese women. Therefore, the rationale to prescribe higher doses of folic acid supplementation has to come from indirect evidence. Several underlying mechanisms have been suggested as determinants in the causal pathway of a relative folate deficiency in obese women, such as chronic inflammation and hyperinsulinemia [15]. However, an overview of causes of folate deficiency in obese women, potential underlying (patho)physiological mechanisms and how they might contribute to a higher risk of NTDs is lacking.

Moreover, different international recommendations on folic acid supplement use for obese women before and during pregnancy are used [16,17]. Therefore, we provide an overview of the epidemiology of folate deficiency in obese (pre)pregnant women, elaborate on potential mechanisms underlying folate deficiency, and discuss considerations for advising higher doses of folic acid supplements. Moreover, we propose suggestions for clinical practice making use of the current evidence, and suggest some areas for further research.

## 2. Epidemiology of Folate Deficiency in Obese (pre)Pregnant Women

### 2.1. Absolute Deficiency

Studies have shown that women with obesity have a lower intake of folate (Table 2). Women with obesity are less likely to use preconceptional folic acid supplement compared to normal weight women, 45.2% versus 60.4%, respectively [12]. They are also less likely to use folic acid supplements on a daily base, 26% versus 33%, respectively [10]. Moreover, women with obesity are less likely to receive enough folate through their diet than lean individuals, i.e., relative malnutrition [18,19]. Both a lower intake of folic acid supplements and a lower dietary intake of folate accounts for lower folate levels in serum, red blood cells, and body fluids. Moreover, decreased folic acid intake is often due to unplanned pregnancies and failed contraceptive methods prevalent in obese women [10].

Though it is clear that obese women have a lower intake of folate, obesity is associated with other factors that are subsequently determinants of a lower intake of folate. Earlier studies indicated that smoking, lifestyle, age, parity, educational level, income level, and whether the pregnancy was planned were determinants of folate intake [20,21]. In a multivariable model, maternal weight status was independently associated with adequate use of folic acid, even after excluding women with an unplanned pregnancy [20].

### 2.2. Relative Deficiency

Obese women had lower serum folate levels even after controlling for folate intake through supplements and diet (β = −0.26, 95% CI: −0.54, 0.02); *p* = 0.07) [22]. When comparing non-obese and obese women with a similar folate intake, serum levels in obese women tend to be lower than in non-obese women, suggesting the current recommendations of folic acid supplement use could be subjected to review.

An increased need for folate is suggested to be caused by altered metabolic processes and chronic low-grade inflammation that could eventually underlie the increased risk for women with obesity on NTDs. Moreover, in women of higher weight categories, an adequate intake of folic acid of 0.4 mg/day did not lower the risk of NTDs [13]. A similar finding was reported by Parker et al., where women with obesity were at increased risk of NTDs, irrespective of adequacy of folic acid intake following the current standard ‘one-fits-all’ dosing regimens [14].

## 3. Theoretical Background

### 3.1. One-Carbon Metabolism

One-carbon metabolism is a complex of interlinking metabolic pathways that are fundamental for molecular biological processes involved in cell multiplication, differentiation, and programming [23]. It provides essential one-carbon moieties used as substrate or cofactor of the linked folate and methionine pathways, as displayed in Figure 1 [24]. We focus on these pathways, however, one carbon metabolism comprises of a series of metabolic pathways [25]. The main substrate of the folate pathway is tetrahydrofolate (THF), which is converted into 5-methyltetrahydrofolate (5-MTHF). Together with homocysteine, it is converted into methionine by methionine synthase (MS) using vitamin B12 as cofactor [26]. The methionine pathway is essential for the provision of methyl groups after transmethylation into S-adenosylmethionine (SAM), the most important methyl donor in the cell [27].

One of the main products of one-carbon metabolism is the contribution to the biosynthesis of nucleotides and epigenetic programming. Interruption of molecular biological processes involved in neural tube development and dependent on one-carbon metabolism, such as cell multiplication, differentiation, apoptosis and programming, can impair the closure of the neural tube. In order to facilitate rapid DNA replication of the tissues involved in the formation of the neural tube, a large pool of nucleotides is required for DNA synthesis and methyl groups for epigenetic programming for neuroepithelial cells. Inadequate supply of nucleotides and methyl groups blocks cellular replication, increases DNA damage, and impairs epigenetic programming and as such the proper development of the neural folds [28]. Given its central role in one-carbon metabolism, folate plays a key role in the molecular biological processes involved in the development of NTDs.

### 3.2. Folate

The most important dietary substrates and cofactors involved in one-carbon metabolism include methionine and choline, together with the B vitamins, cobalamin, and folate. Folate is an essential water-soluble B-vitamin and naturally occurs in fruits and vegetables. Folate-rich foods include in particular leafy green vegetables, lentils, beans, and citrus fruits. In general, the term folate refers to the natural forms in foods and body fluids, while the term folic acid applies to the more stable but synthetic supplemental form. Folate is a crucial mediator in the one-carbon metabolism, where it acts as a dietary methyl donor together with methionine, betaine, and choline [23]. Folate derived from food needs to be hydrolysed from polyglutamates to monoglutamates, before absorption takes place in the jejunum [29]. This process leads to a lower bioavailability that varies between 30% and 98% [30,31]. Another source is synthetic folic acid, present in fortified foods and in various supplements. The bioavailability of this form is commonly estimated at 85% [32]. Synthetic folic acid is a monoglutamate and needs to be converted by dihydrofolate reductase (DHFR) to be taken up in its the active form, THF, in the intestinal cells.

### 3.3. Epigenetics

Generally, epigenetics is defined as the alterations in the gene expression profile of a cell that are not caused by changes in the DNA sequence [33]. Epigenetics is critical to normal genome regulation and development. One-carbon metabolism is essential for epigenetic modifications by providing methyl groups for the methylation of DNA and associated (histone) proteins as well as RNA, for which an adequate folate supply is important. With one-carbon metabolism being essential, it is plausible that folic acid plays a role in epigenetics and its related plasticity of gene methylation. Indeed, periconceptional folic acid supplement use has been shown to be associated with epigenetic changes [34]. Although, maternal intake of folic acid supplements and dietary folate are positively associated with long interspersed nuclear elements (LINE-1) methylation, a surrogate marker of global DNA methylation, transgenerational effects could not be demonstrated in cord blood [35,36,37].

Such epigenetic modifications, particularly where DNA methylation is involved, have been proposed as plausible mechanisms underlying associations between folate and various disease outcomes, such NTDs, cardiovascular diseases, and cancer [38].

## 4. Pathophysiology of Relative Deficiency of Folate in Obese Women

### 4.1. Impaired One-Carbon Metabolism

Hyperhomocysteinemia, conventionally described as a serum level above 15 micromol/L, is a sensitive marker of an impaired one-carbon metabolism [39]. Considering the pathways within the one-carbon metabolism, a folate deficiency and as such less supply of methyl groups, contributes to higher levels of homocysteine, and higher levels of homocysteine lead to a higher demand for folate used for remethylation of homocysteine [40]. Moreover, hyperhomocysteinemia is a risk factor for several poor health outcomes, including, among others, neurological disorders, vascular diseases and reproductive disorders [23,41,42]. Pregnancy complications such as preeclampsia, intra-uterine growth restriction, and prematurity are associated with high maternal levels of homocysteine [23,43,44]. Hyperhomocysteinemia is more common in women with obesity, compared to non-obese individuals: two studies reported statistically significant differences in homocysteine levels between obese and non-obese women; 12.76 ± 5.30 μM/L versus 10.67 ± 2.50 μM/L, respectively, and 10.2 μM/L [4.6–26.3] versus 8.9 [4.4–25.8] respectively [45,46]. Suggested folate-related pathways that could underlie this finding are discussed below. In addition, an overview of potential underlying (patho)physiological pathways of folate deficiency and NTDs in obese women is displayed in Figure 2.

### 4.2. Physiology of Adipocytes

Adipose tissue is traditionally categorized into white and brown adipose tissue. Brown adipose tissue is specialized in energy expenditure and thermogenesis [47,48]. White adipose tissue is responsible for storing and releasing energy in the human body by controlling lipogenesis and lipolysis, respectively. During the process of lipogenesis, free fatty acids and glycerol are taken up from the blood stream and are stored as triglycerides in adipocytes [49]. On the contrary, lipolysis is the mechanism by which triglycerides are catabolized into free fatty acids and glycerol that are released into the bloodstream where they act as an energy source for other organs [50].

Obesity is characterized as an excessive growth of adipose tissue [51]. Furthermore, obesity is known to cause both hypertrophy as hyperplasia of the adipocyte [52]. These processes are associated with an infiltration of macrophages into the adipose tissue. This promotes inflammation and introduces TNFα into the tissue [53]. Moreover, the expansion of adipose tissue in obesity is linked to an inappropriate supply with oxygen and hypoxia development [54]. Subsequent inflammatory reactions inhibit preadipocyte differentiation and initiate adipose tissue fibrosis [55]. Not all obese individuals develop adipose tissue fibrosis followed by inflammation; however, obesity-related hypertrophic adipocytes may induce inflammation by producing pro-inflammatory adipokines [56].

### 4.3. Pro-Inflammatory State

The obesity-related low-grade chronic inflammation is generated by the production of pro-inflammatory cytokines, as IL-6 and TNF-α, and adipokines, as leptin [57]. Consumption of excess energy may as well acutely induce inflammatory responses [58,59]. Hence, it is thought that excess energy by overfeeding is another starting signal of inflammation, causing overactivation of tissues involved in metabolism, like adipose tissue, liver, and muscle, which in reaction to this stimulus provokes the inflammatory response [60,61]. Thus, besides continuous, low-grade chronic inflammation, there also might be additional, acutely induced inflammatory responses caused by excess supply of food. The inflammation-related collateral tissue damage activates tissue repair responses, requiring one-carbon moieties for synthesis of adequate amounts of proteins, lipids, nucleotides, and others. Since the folate dependent one-carbon metabolism supports cell proliferation at the detriment of B-vitamins, obesity-induced inflammation is associated with hyperhomocysteinemia and, thereby, folate deficiency. In addition, hyperhomocysteinemia is not only a result of inflammation, but hyperhomocysteinemia will again promote inflammation due to the excessive oxidative stress generated from high homocysteine levels [62].

### 4.4. Insulin Resistance

Adipose tissue regulates energy storage and release by lipogenesis and lipolysis. Obesity is associated with an increased basal lipolysis, which might be caused by an impaired sensitivity of adipocytes to insulin signaling, overexpression of the leptin gene in adipocytes, and increased circulating levels of leptin [63]. By the increased rate of lipolysis, higher amounts of fatty acids and glycerol are catabolized and enter the bloodstream. Increased serum levels of fatty acids, non-esterified fatty acids (NEFAs) in particular, are considered to be the most critical factor in inducing insulin resistance [50]. This is a pathological condition in which the capacity of cells to respond to normal levels of insulin is reduced. Increased NEFA levels are observed in persons with obesity and are associated with insulin resistance. Moreover, insulin resistance establishes within hours after an acute increase in plasma NEFA levels [64]. Beside the lipolysis-derived factors, the increased release of inflammatory cytokines influences the development of insulin resistance as well [65,66]. Especially, TNF-α and IL-6 cause an upregulation of potential mediators of inflammation that contribute to insulin resistance.

Additionally, chronic inflammation in general is not only associated with hyperhomocysteinemia and folate deficiency, but also with insulin resistance [67]. Although the exact working mechanism is not unravelled yet, it is suggested that insulin resistance influences activity of key enzymes in the folate dependent one-carbon metabolism, including 5,10-methylenetetrahydrofolate reductase (MTHFR) and cystathione b-synthase (CBS) [68,69]. Furthermore, it has been demonstrated that insulin signaling is affected by high levels of homocysteine, which is a condition associated with obesity [70,71]. Insulin signaling is an essential process in glucose homeostasis, since it increases the uptake of glucose into muscle and fat cells and reduces the synthesis of glucose in the liver. GLUT4 is one of the most important insulin-regulated glucose transporters responsible for decreasing blood glucose concentrations by facilitating glucose uptake into muscle and adipose tissue [72]. In the absence of insulin, the majority of GLUT4 is sequestered in intracellular vesicles in muscle and fat cells. When insulin levels increase, translocation of GLUT4 to the plasma membrane is induced and diffusion of circulating glucose down its concentration gradient into muscle and fat cells is facilitated. Homocysteine is one of the factors known to disrupt insulin signaling by impeding the GLUT4 translocation or recruitment on the plasma membrane and therefore reducing glucose uptake, which results in higher levels of glucose in the blood plasma [67].

### 4.5. Hyperglycaemia

Insulin resistance forces the pancreatic β-cells to produce more insulin to be able to prevent hyperglycaemia. However, when the compensatory insulin production is no longer sufficient, excessive amounts of glucose circulate in the blood plasma. This condition is referred to as hyperglycaemia, which is a defining characteristic of diabetes mellitus [73]. Besides maternal obesity, diabetes mellitus is a known risk factor for NTDs. Both obesity and diabetes mellitus are features of the metabolic syndrome [15]. The metabolic syndrome is further characterized by other metabolic risk factors including dyslipidemia, chronic hypertension, proinflammatory state, and prothrombotic state [74]. In the presence of 1 or 2 features of the metabolic syndrome, the fetus is on a 2-fold and 6-fold higher risk for NTD, respectively [75]. While the increased risk of NTDs associated with obesity appears to be independent of diabetes, a possible mechanism might be hyperglycemia due to insulin resistance in obese women [15].

Glucose levels are monitored and regulated by the islets of Langerhans in the pancreas and glucose is an essential factor for aerobic metabolism. Evidence suggests that the early developing embryo is dependent on maternal glucose metabolism, with detrimental effects in case of disbalance and hyperglycemia [76]. Thus, at the time of neural tube closure (around the fourth week of gestation), mothers with poorly regulated glucose levels are likely to have an suboptimal in utero environment, causing abnormal organogenesis [43,77,78]. To date, the exact working mechanism has not been elucidated yet. Only a few studies have reported evidence for this explanation, mostly focusing on the genetic susceptibility related to hyperglycemia as a risk factor for NTDs. Previous animal studies investigating molecular causes of NTDs in the embryos of diabetic mothers, demonstrated that in mouse embryos, expression of Pax3 is suppressed beginning on embryonic day 8.5 and subsequently, neuroepithelial cells undergo apoptosis and NTDs occur at increased frequency compared to embryos from nondiabetic pregnancies [79]. Moreover, in an embryos mouse model, which demonstrates a homozygous loss of function mutation in the Pax3 gene, NTDs can be rescued by either folic acid or thymidine supplementation [80,81]. This finding suggests that folic acid prevents NTDs by ensuring sufficient biosynthesis of factors for cell proliferation. Furthermore, a recent review of randomized controlled trials indicated that folic acid supplementation in non-pregnant populations, including women and men, had potential benefits on insulin resistance and glycemic control [82]. The mechanisms by which folic acid supplements lowers glucose levels and insulin resistance are still unclear. One of the suggested explanations is that hyperhomocysteinemia increases vascular oxidative stress, which could relate to insulin resistance and impaired insulin secretion during hyperglycemia [83,84]. As such, folate or folic acid supplements might decrease oxidative stress and, thereby, could prevent hyperglycemia and its detrimental effects.

### 4.6. Inositol

Inositol has been the focus of a large number of studies and is also involved in both folate uptake and glucose metabolism. Myo-inositol and D-chiro inositol are inositol isomers. Myo-inositol is the predominant form, which can be produced by the human body from D-glucose and is naturally present in foods, such as cereals, legumes, and meat [85]. Both isomeric forms of inositol were found to have insulin-like properties, acting as second messengers in the insulin intracellular pathway. Furthermore, both of these molecules are involved in increasing insulin sensitivity of different tissues, and thereby, improving health outcomes associated with insulin resistant, such as diabetes mellitus and reproductive disorders [86,87,88]. A randomized controlled trial showed that myo-inositol supplementation, started in the first trimester, in obese pregnant women reduced the incidence of gestational diabetes mellitus in the myo-inositol group compared with the control group, 14.0% compared with 33.6%, respectively (*p* = 0.001; odds ratio 0.34, 95% confidence interval 0.17–0.68) [89]. This reduction was achieved by improving insulin sensitivity.

Besides the insulin-like properties, an animal study demonstrated that myo-inositol is capable of significantly reducing the incidence of spinal NTDs in curly tail mice, a genetic model of folate-resistant NTDs [90]. Furthermore, in humans, significantly lower inositol concentrations have been reported in the blood of mothers carrying NTD fetuses compared with normal pregnancies, and mothers with low blood levels of inositol showed a 2.6-fold increased risk of an affected offspring [91].

Moreover, inositol is suggested to have preventive effects on NTD occurrence in curly tail mutant mouse [90]. Protection against diabetes-induced NTDs has been observed as well in other rodent models [92]. Hence, the animal data support a distinct inositol-dependent metabolic pathway that, when stimulated, can prevent NTDs.

### 4.7. Role of the Gut Microbiome

The gut microbiome can directly influence the folate status and via the cofactors vitamin B12 en B2, which contribute to a relative folate deficiency. The gut microbiome is the entirety of microorganisms, bacteria, viruses, protozoa, and fungi, and their collective genetic material present in the gastrointestinal tract [93]. For this overview, we focus on the bacterial microbiome. Gut bacterial microbiota are involved in a variety of essential processes, including the fermentation of indigestible food components into absorbable metabolites, the synthesis of essential vitamins, such as folate and vitamin B12, the removal of toxic compounds, the strengthening of the intestinal barrier, and the stimulation and regulation of the immune system [94,95,96]. Diversity is of great importance to a healthy intestinal microbiome, since it ensures redundancy, with multiple microbes competent to perform similar functions [97]. An imbalance in microbial populations, called dysbiosis, is associated with several poor health outcomes, including, among others, inflammatory bowel disease, neurological diseases, and diabetes [98,99]. Moreover, there is increasing evidence, mainly from animal studies, that alterations in the intestinal microbiome lead to metabolic and weight changes in the host [100,101].

An animal study found in genetically obese mice a 50% reduction in the abundance of Bacteroidetes and a proportional increase in Firmicutes [102]. Moreover, it is noted that changes affect the metabolic potential of the mouse gut microbiota. Previous research indicated that the obese microbiome has an increased capacity to harvest energy from the diet [101]. Furthermore, this trait is transmissible: colonization of germ-free mice with an ‘obese microbiota’ results in a significantly greater increase in total body fat than colonization with a ‘lean microbiota’. Besides the role of the gut microbiota as a contributing factor to the pathophysiology of obesity, it is also recognized as a source of B vitamins, in particular of folate and vitamin B12. It is produced by the colonic microbiota, mainly as the monoglutamate form of folate, the form that is absorbed at the highest rate. Thus, intestinal bacteria are a source of folate [103]. Even though absorption of folate occurs primarily in the duodenum and upper jejunum, the colon represents another depot of folate potentially affecting the general folate status of the host.

Moreover, the composition of the intestinal microbiome contributes to the regulation of intestinal permeability [104]. Short-chain fatty acids have been suggested as a mediator via which intestinal microbiota might promote the integrity of the intestinal mucosa. A higher intestinal permeability has been associated with obesity, leading to a ‘leaky gut’ with suboptimal uptake of micronutrients [105]. Hypothetically, there might be a derangement in the absorption of folate as well.

## 5. Considerations for Advising Higher Doses of Folic Acid Supplements

Positive effects of folic acid supplement use on NTD birth prevalence rates in the general population are shown in doses ranging from 0.36 mg (NTD occurrences) to 4 mg (NTD recurrences) per day. However, after these randomized controlled trials, further investigation into an optimal dose for preventive effects could not be performed anymore due to ethical considerations [9,106,107,108].

The presence of unmetabolized folic acid, which accumulates in serum above doses of 0.2 mg per day, is generally regarded as a marker of dihydrofolate reductase (DHFR) saturation in its capacity to convert folic acid to tetrahydrofolate (THF) [109,110,111,112].

Various animal experiments showed that folic acid, especially when applied directly into the brain, possess powerful excitatory and convulsive properties by unknown mechanisms, although evidence suggests that unmetabolized folic acid might induce neurotoxicity [113,114,115].

An observational study reported an increased risk of impaired psychomotor development with the use of 5 mg of folic acid per day [116]. Daily intakes of 800 µg to 5 mg of folic acid from supplements have been associated with an increased risk of cancer development and mortality perinatally and later in life [117]. Since folate is an important methyldonor for periconceptional epigenetic programming, high doses of folic acid can induce variations in the epigenome of the offspring [34,118]. Until now, there is no conclusive evidence which dose of folic acid supplement use causes adverse effects in either the pregnant woman or the fetus [119].

There is only indirect evidence that obese women could benefit from an increased dose of folic acid in the prevention of NTDs in the offspring, as discussed in the previous sections. Hence, until the possible alterations in folate metabolism and corresponding requirements of folic acid supplement use in obese women are clarified, an increased folic acid supplementation dosage is only justified when harmful effects are ruled out.

## 6. Current Guidelines

In the previous sections, we described plausible folate-related pathways underlying the increased risk of NTDs in the offspring of obese women. No study has performed a trial where obese women are randomized to a high dosage versus a normal low dosage, and are followed-up until birth outcomes, including NTDs. As both a relative folic acid deficiency and insulin resistance are plausible mechanisms, direct evidence that an increased dosage of folic acid prevents NTDs in obese women is lacking [120]. Therefore, current guidelines are based on indirect evidence, which may explain the differences in these guidelines. British and Australian guidelines recommend 5 mg/day of folic acid in obese women, while American and Canadian guidelines do not mention special recommendations for folic acid supplement use in obese women [16,17]. These differences in recommended folic acid supplement use for obese women might be related to national folic acid food fortification programs. In the United States and Canada, folic acid fortification of most cereal grains is mandatory, while in the United Kingdom and Australia, this is only applied to wheat flour. New guidelines should not only be based on substantial scientific evidence. Local or national circumstances or customs, such as folic acid food fortification programs, should also be taken into account.

## 7. Recommendations

### 7.1. Recommendations for Practice

Although there is insufficient evidence that it is effective and safe to increase the recommended dose of folic acid supplement use for obese (pre)pregnant women in the prevention of neural tube defects, we formulated the following recommendations for clinical practice to improve absolute folate deficiency, either through supplement use or dietary intake:Be aware of a suboptimal absolute folate intake in obese women, both as a result of a lack of compliance to folic acid supplement use as well as of a relative malnutrition due to a folate deficient diet, as discussed in Section 2. More than half of pregnant women reported to start using folic acid supplements after a positive pregnancy test, which is on average after 5.5 weeks of gestation [121,122]. Since the closing of the neural tube occurs between week 4 and 6 of pregnancy, the majority of pregnant women start using folic acid supplements too late for the prevention of NTDs (Figure 3). Therefore, the preconception period is the window of opportunity to determine and treat folate deficiency or hyperhomocysteinemia in women with obesity and provide lifestyle counseling to improve dietary folate intake and stimulate weight loss [123]. Additionally, parameters of chronic inflammation and glucose metabolism could be measured as a risk analysis. Face-to-face lifestyle counseling could be combined with an online program, for example the evidence-based eHealth platform ‘Smarter Pregnancy’. This eHealth intervention showed improvements in lifestyle behaviors, including folic acid supplement use and nutritional intake, in the total study population as well as in the subgroup of overweight and obese women [124]. Since unplanned pregnancies and failed contraceptive methods are prevalent in obese women, this group is less likely to attend preconception care. As presented in Figure 3, folic acid supplement use in general should start before conception to have its full potential. Therefore, the general practitioner could inform women, independent of their BMI, who, for example, stop taking their contraceptives.Obese women can be monitored by assessment of serum folate and red blood cell folate during the periconceptional period, as well as plasma total homocysteine status. Based on these parameters, folate status, one-carbon metabolism, and related pathways can be improved by supplements or lifestyle counseling, the latter being preferred because of no concerns about safety.

### 7.2. Recommendations for Future Research

A preconceptional initiated intervention study to explore the etiology of insulin resistance and chronic inflammation in obese women and the effects of increased folic acid supplement use.Modification of the intestinal microbiota to maintain intestinal permeability and adequate uptake and production of essential nutrients is worth further research.Further research should focus on the implementation of interventions to target absolute folate deficiencies. Lifestyle programs have the potential to increase dietary folate intake, folic acid supplement use, and overall lifestyle improvement among obese women [124]. Wide implementation and evaluation of such interventions could provide a powerful preventive measure.

## 8. Conclusions

Women with obesity are at an increased risk of NTDs in their offspring and there is substantial evidence that folate deficiency plays a significant role. However, clinical trials to show the optimal dose of folic acid supplement use are lacking. Scientific evidence of the involvement of several folate-related pathways implies to increase the recommended folic acid supplement use in obese women. However, the physiological uptake of synthetic folic acid is limited and side-effects in mothers and offspring, in particular variations in epigenetic (re)programming with long-term health effects, cannot be excluded. Therefore, we emphasize the urgent need for preconception personalized counseling on folate status, lifestyle and medical conditions, in particular for women with obesity. Targets for further research to substantiate folic acid recommendations in women with obesity are directed towards homocysteine, glycemic control, and the microbiome. We recommend that folic acid supplement use guidelines should be reconsidered when more scientific evidence is available.

## Figures and Tables

**Figure 1 nutrients-13-00331-f001:**
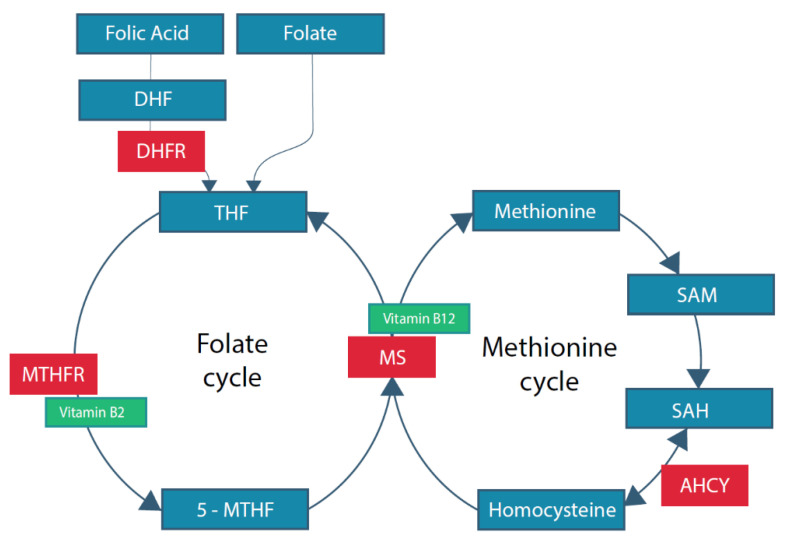
Folate related one-carbon metabolism. DHF: dihydrofolate; DHFR: dihydrofolate reductase; THF: tetrahydrofolate; MTHFR: methylene tetrahydrofolate reductase; 5-MTHF: 5-methyltetrahydrofolaat; MS: methionine synthase; SAM: S-adenosyl-methionine; SAH: S-adenosyl-homocysteine; AHCY: S-adenosylhomocysteine hydrolase.

**Figure 2 nutrients-13-00331-f002:**
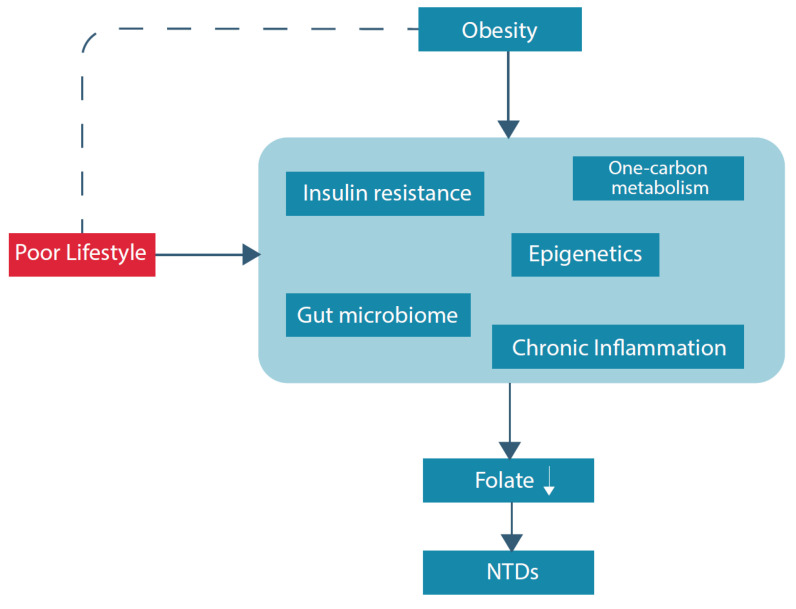
Overview potential underlying (patho)physiological pathways of folate deficiency and NTDs in obese women.

**Figure 3 nutrients-13-00331-f003:**
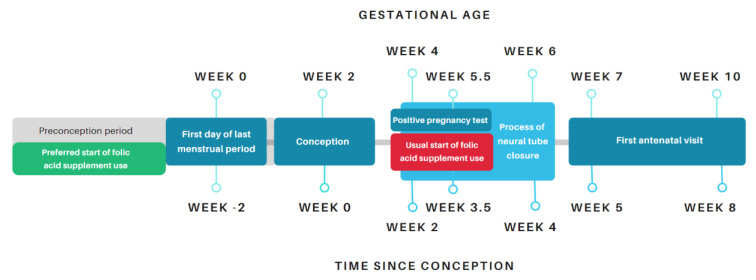
Illustration of the gap between recommended period of folic acid supplement use, and window of opportunity for the health care provider to advice on folic acid supplement use.

**Table 1 nutrients-13-00331-t001:** Overview of three meta-analyses on the association between maternal obesity and NTD in offsprings.

	Years Included	Number of Studies	Design	Results (OR (95% CI))			
				Normal Weight	Overweight	Obese	Severely Obese
**Rasmussen et al. 2008 [4]**	January 2000–January 2007	12	Cohort and case-control studies	1 (ref)	1.22 (0.99–1.49)	1.70 (1.34–2.15)	3.11 (1.75–5.46)
**Stothard et al. 2009 [5]**	January 1966–May 2008	18	Cohort and case-control studies	1 (ref)		1.87 (1.62–2.15)	
**Huang et al. 2017 [6]**	up to 15 December 2015	22	Case-control studies	1 (ref)	1.20 (1.04–1.38)	1.68 (1.51–1.87)	

**Table 2 nutrients-13-00331-t002:** Intake of folate and folic acid supplements in women, per weight category.

	Study Design	Population	Sample Size	Outcome	Results (% or Mean ± SD)			
					Normal Weight	Overweight	Obese	*p*-Value
**Masho et al. 2016 [10]**	Cohort study	Women with singleton pregnancy living in USA	104.211	Daily intake of folic acid supplement	33%	29%	26%	<0.0001
**Farah et al. 2013 [12]**	Cohort study	White European women with a singleton pregnancy	288	Use of folic acid supplement	60%	60%	45%	0.029
**Bird et al. 2015 [18]**	Cohort study	Non-pregnant women aged ≥19 years living in the USA	538	Folate intake through diet (μg/L)	559 ± 12.7	557 ± 14.5	517 ± 10.5	0.002

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
