# Peer review of "Epidemiology and (Patho)Physiology of Folic Acid Supplement Use in Obese Women before and during Pregnancy"

_nutrients, 2021, doi:10.3390/nu13020331_

Round 1
Reviewer 1 Report
This review is well written in a theoretical and clinically applicable form.
The topic of the present study is relevant and interesting
It adds a piece of evidence to this area of knowledge.
The paper is well written, clear and easy to read.
The conclusions are consistent with the presented arguments and address the main question posed.
Author Response
Thank you for your positive feedback on our manuscript.
Reviewer 2 Report
Nutrients
The review article focuses on an important public health topic, folate supplementation in obese pregnant women.
The strengths of the study are the review of studies conducted on humans and animal models.
The weak points of the review are that they have not discussed the heterogeneity of public health measures in the different continents and the differences between countries where there is a systematic fortification in folates.
- Epidemiology of folate deficiency in obese (pre)pregnant women
Line 96: Table 2 For Bird et al., it is not clear the mean. Is-it correspond to folate intake ? Could you clarify?
- Theoretical background
Lines
147 – 149. Another source is synthetic folic acid, present in fortified foods and in various supplements. The bioavailability of this form is commonly estimated at 85%
- Pathophysiology of relative deficiency of folate in obese women
Line 173. Hyperhomocysteinemia is also related to pregnancy complications other than NTD (repeated miscarriage, pre-eclampsia, placenta abruptio, thromboembolic events, and perhaps with fetal death-in-utero and intra-uterine growth retardation). Please adds this in your manuscript
Line 209. Something is missing in your sentence (Underlined in yellow): “The inflammation-related collateral tissue damage activates tissue repair responses”. ,“requiring one-carbon metabolism to provide one-carbon moieties for synthesis of sufficient proteins, lipids, nucleotides and others”.
- Current guidelines
Lines 365-367
The authors comment that in Britain and Australia they recommend a 5 mg folate supplement for obese pregnant women. And that in the United States and Canada these recommendations do not exist. I think the authors should discuss why these differences. Probably the difference is the following: Britain and Australia are countries where there is no systematic fortification in folates, and in countries such as Canada and the United States, foods are systematically fortified.
Lines 395-399
The WHO has proposed in 2015 certain guidelines to determine the status of folates. Authors should cite these universal propositions.
Author Response
Please see the attachment

This manuscript is a resubmission of an earlier submission. The following is a list of the peer review reports and author responses from that submission.
Round 1
Reviewer 1 Report
The present review provides an overview of epidemiologic evidence of the use of different doses of folic acid in association with the prevention of neural tube defects, and elaborates on potential mechanisms underlying folate deficiency in obese women.
Rationale
Please consider to add also the following reference for the definition of obesity as BMI>30 Kg/m2: Eveleth, P.B.; Andres, R.; Chumlea, W.C.; Eiben, O.; Ge, K.; Harris, T.; Heymsfield, S.B.; Launer, L.J.; Rosenberg, I.H.; Solomons, N.W.; et al. Uses and interpretation of anthropometry in the elderly for the assessment of physical status. Report to the Nutrition Unit of theWorld Health Organization: The Expert subcommittee on the Use and Interpretation of Anthropometry in the Elderly. J. Nutr. Health Aging 1998, 2, 5–17.
Moreover, I suggest to classify women also for their gestational weight gain (GWG) during pregnancy. Please, consider the following: DOI:10.3390/nu11061308. Are there any studies that evaluate the different recommendations for folic acid supplement use considering GWG?
Epidemiology of folate deficiency in obese (pre)pregnant women
In this section, the authors reported several results based on different studies with different studies population. For this reason, I suggest to give more details on study design and population when the authors provide specific results (percentages, OR..).
In my opinion, the authors could add a table giving more details and results about the above-mentioned studies.
Moreover, I suggest to expand the paragraph taking into account other results about folic acid supplement among pregnant women. Please, consider the following: DOI: 10.3390/ijerph17020638.
Theoretical background
For the paragraph “Epigenetics”, the authors should expand the paragraph taking into account other factors that could influence the methylation process. Please, consider the following:
DOI: 10.1038/s41598-020-71352-9.
DOI: 10.3390/medicina56080374
DOI: 10.3390/nu11081843
DOI: 10.1007/s12263-015-0480-4
DOI: 10.1371/journal.pone.0109478
Pathophysiology of relative deficiency of folate in obese women
Why Figure 2 is cited in this section, and in particular in the paragraph “Role of the gut microbiome”?
Reviewer 2 Report
The authors of this narrative review explore the evidence for folic acid supplementation in pregnancy in women with obesity. The authors do an excellent job of succinctly summarizing many different areas of science that may play a role in the pathophysiology of NTD.
While the authors build their argument for higher doses of folic acid in women with obesity, they do not spend time reviewing the NTD upon which the current recommendations are derived. There is extremely limited data demonstrating the efficacy of folic acid supplementation above 1 mg in preventing NTD. Further, studies in animals and humans have shown that at doses higher than 1 mg there is not increased absorption and instead excretion of excess folate through the urine. At high doses unmetabolized folic acid can be found in the blood.
Reviewer 3 Report
There is a known higher risk of NTD pregnancy among obese women, however, there is, as yet, no metabolic or clinical evidence that this risk can be reduced by folic acid supplementation. Nevertheless, because of this risk, some health agencies recommend that obese women take folic acid supplements above the general recommended amount pre and during pregnancy. There is considerable interest and an active literature researching the topic.
As a general comment I feel that this review fails to live up to the views expressed in the title or in the abstract and the narrative needs considerable revision throughout to provide a focused critique on the topic. Lines 67-73 of the abstract give an important guide to what the review is claiming to discuss. The section states ‘we provide an overview of the epidemiologic evidence of the use of different doses of folic acid supplementation in association with the prevention of NTDs in the offspring and put our findings in perspective given the different recommendations in current international guidelines’. I don’t quite know what the authors are suggesting here but I cannot see any actual discussion or consideration of these points in the text. The following sentence … ‘we address potential pathways, suggest diagnostic and treatment options… etc’ is even more aspirational but again is not discussed adequately in the text. Sections 6 and 7, relating to current guidelines and recommendations, are presented as stand-alone sections that are drawn from common knowledge rather than a synthesis of the arguments presented in sections 2 to 5. While these sections are relevant to public health, they do not tie into any aspect of the review narrative. Overall the review needs revision to provide a sense of purpose and continuity.
The review begins by summarizing the topic of NTD risk among obese mothers in a few lines. However, without any in-depth analysis of the literature to date on this topic, the arguments put forward in the review are not well rooted in the background. I would like to see more detail on studies on maternal obesity in relation to pregnancy and neural tube defects.
The review then divides into two general discussions; the first is a relatively brief overview on folate metabolism (sections 2 and 3) followed by a much more in-depth and informative discussion of the physiological and metabolic changes observed in in obesity (section 4).
Section 2.1: The authors present an interesting discussion of absolute and relative folate deficiency; however, one important reason for low periconceptional folic acid supplement use is unintended pregnancy. This is not considered by the authors but is widely discussed in the literature in relation to obesity and has important relevance in relation to whether recommendations for folic acid supplement dosage should be increased in obese women as opposed to ensuring that general recommendations are simply followed. In my opinion, this background is highly relevant to the current review and should be included in the discussion.
Line 85 (minor point): The sentence ‘This finding makes the presence of a relative folate deficiency, caused by a higher consumption of folate, in obese women plausible.’ does not make sense. Please clarify.
Sections 3.1 and 3.2 on folate and one-carbon metabolism pathways do not demonstrate a clear understanding of this area. It is not correct to imply that dietary folate enters the pathway through a mechanism that is independent of vitamin B12 dependent methionine synthase, and lines 131-132 are factually incorrect, i.e. dietary folate is absorbed as MTHF monoglutamate, not as THF polyglutamate. Furthermore, Fig 1 gives no indication of how one-carbon units enter the system and the entire cycle that provides one-carbon units for DNA synthesis is omitted, yet this is an important part of the narrative between lines 105 and 114. In contrast, homocysteine catabolism to taurine and sulfate – a pathway beyond folate metabolism and strongly expressed only in a few tissues , is featured in detail. This entire section requires revision to attain a reasonable standard of accuracy for a lay reader.
In Section 4 the authors present an in-depth description of metabolic and physiolofgical changes in adipose and other tissues. This section is informative although several key statements need to be referenced. For example the sentence in Lines 171- 172 on expansion of adipose tissue should be referenced. The review would be greatly improved by a more in-depth discussion on the role of one-carbon metabolism in obesity. There is also no evaluation or argument to propose or dispute alternative recommendations for folic acid supplement use for obese women in relation to pregnancy. This is a stated objective of the review.
The flow of argument in lines 186-190 is unclear, and some statements (eg lines 186-187) do not make sense. Please revise.
Section 5 is not relevant to the topic and could be removed.
Figure 3 is unnecessary and should be omitted.
The manuscript has not been properly checked prior to upload, at least in relation to references. As a reviewer, I regard this as disrespectful to the Journal and to people who are asked to carry out the peer-review. By reference 20 in the downloaded article, I stopped trying to work out what references were actually associated with citations. Please fix these.
Throughout the manuscript, there are syntax and grammatical errors that should be fixed.
Round 2
Reviewer 1 Report
The authors addressed all my comments and suggestions.
Reviewer 2 Report
Approve of revised version